# Synthetic phosphoethanolamine-modified oligosaccharides reveal the importance of glycan length and substitution in biofilm-inspired assemblies

Theodore Tyrikos-Ergas[1,2,4], Soeun Gim[1,2,4], Jhih-Yi Huang [1,2,4], Sandra Pinzón Martín[1,2], Daniel Varón Silva [2,3], Peter H. Seeberger[1,2] & Martina Delbianco [1✉]

Bacterial biofilm matrices are nanocomposites of proteins and polysaccharides with remarkable mechanical properties. Efforts understanding and tuning the protein component have been extensive, whereas the polysaccharide part remained mostly overlooked. The discovery of phosphoethanolamine (pEtN) modified cellulose in *E. coli* biofilms revealed that polysaccharide functionalization alters the biofilm properties. To date, the pattern of pEtN cellulose and its mode of interactions with proteins remains elusive. Herein, we report a model system based on synthetic epitomes to explore the role of pEtN in biofilm-inspired assemblies. Nine pEtN-modified oligosaccharides were synthesized with full control over the length, degree and pattern of pEtN substitution. The oligomers were co-assembled with a representative peptide, triggering the formation of fibers in a length dependent manner. We discovered that the pEtN pattern modulates the adhesion of biofilm-inspired matrices, while the peptide component controls its stiffness. Unnatural oligosaccharides tune or disrupt the assembly morphology, revealing interesting targets for polysaccharide engineering to develop tunable bio-inspired materials.

[1] Department of Biomolecular Systems, Max Planck Institute of Colloids and Interfaces, Am Mühlenberg 1, 14476 Potsdam, Germany. [2] Department of Chemistry and Biochemistry, Freie Universität Berlin, Arnimallee 22, 14195 Berlin, Germany. [3] Present address: Institute of Chemistry and Bioanalytics, School of Life Sciences, University of Applied Sciences and Arts Northwestern Switzerland, Hofackerstrasse 30, 4132 Muttenz, Switzerland. [4] These authors contributed equally: Theodore Tyrikos-Ergas, Soeun Gim, Jhih-Yi Huang. ✉email: martina.delbianco@mpikg.mpg.de

**B**acteria secrete various biomolecules to create extensive networks of extracellular matrix (ECM). These biofilms, often associated with pathogenic infections[1], have gained popularities for their remarkable mechanical properties, transforming bacteria into elegant biofactories of smart materials[2-5]. The major components of the ECM of *Escherichia coli* (*E. coli*) biofilms are curli fibrils—bacterial functional amyloids[6,7]—and cellulose[8,9] (Fig. 1a). Genetic engineering approaches[10] permitted the programming of bacterial amyloid production[11,12] to generate tunable bioplastics[13]. Similar strategies to tune the production of bacterial polysaccharides[4,14,15], the other major components of bacteria biofilms, are limited by complex biosynthetic pathways.

Recently, it was discovered that some bacteria (e.g., *E. coli* and *Salmonella enterica*) produce chemically modified cellulose bearing phosphoethanolamine (pEtN) substituents[16]. The

composite of curli and pEtN cellulose generates biofilms with enhanced elasticity and adhesion to bladder epithelial cells[17]. This exciting discovery suggests that the carbohydrate component tunes the biofilm properties and may be the basis for tailoring cellulosic materials for applications in tissue engineering, biotechnology, and the food industry[18,19]. With the genes responsible for the pEtN modification identified[16], genetically engineered bacteria could be imagined for the production of specifically modified cellulose[14,20-22].

Several fundamental aspects remain to be elucidated before pEtN cellulose can be exploited to its full potential. Approximately half of the glucose units of cellulose are substituted at the C-6 hydroxyl group with a pEtN moiety and the ratio of curli-to-pEtN cellulose varies among different *E. coli* strains[23]. The pattern of modification, the length of the pEtN cellulose, and the

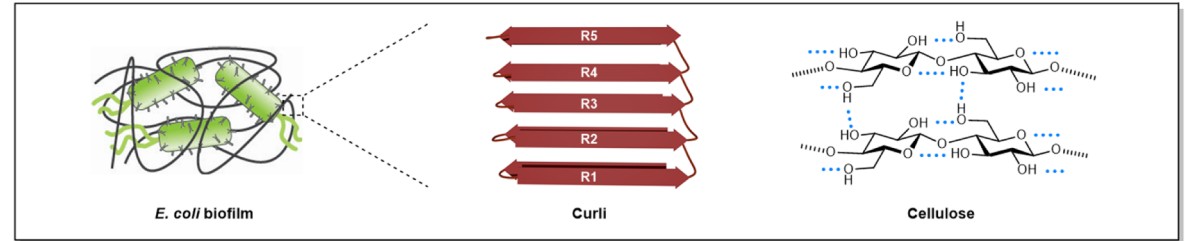

**b** Representative synthesis of (PA)₃

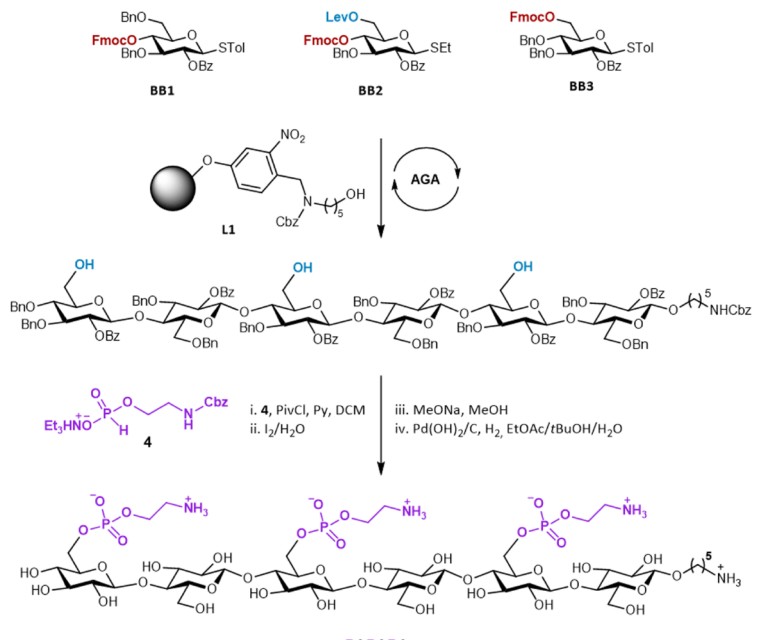

**PAPAPA**

**c** Synthetic oligosaccharides

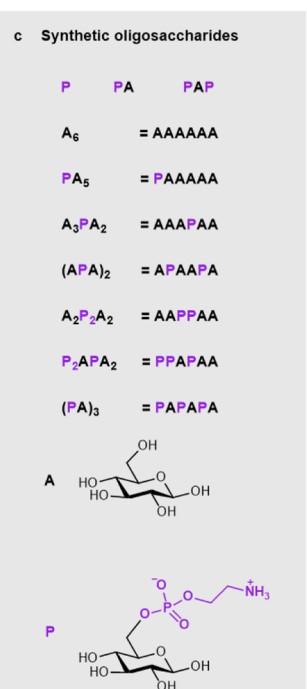

**d** Chemical structure of R5

R5 =   S   S   V   N   V   T   Q   V   G   F   G   N   N   A   T   A   H   Q   Y

**Fig. 1 *E. coli* biofilm and synthesis of its representative matrix components. a** Cartoon representation of *E. coli* biofilm. **b** Representative synthesis of a pEtN hexasaccharide. AGA includes cycles of glycosylation, capping, and Fmoc deprotection. A final Lev deprotection liberates the hydroxyl groups that are functionalized in post-AGA steps. Reaction conditions for AGA are reported in the SI. PivCl = pivaloyl chloride, py = pyridine, MeONa = sodium methoxide. **c** Collection of oligosaccharides synthesized in this work. **d** Chemical structure of the peptide **R5**.

mode of interaction with curli remain unknown[18]. Pure, well-defined oligosaccharide standards are essentials to better understand pEtN cellulose and its role in the ECM, in anticipation of applications. Isolation of pEtN cellulose from natural sources generates ill-defined mixtures and may alter its chemical structure[17]. Chemical synthesis can provide standards with precise control over the sequence, length, and substitution pattern[24,25]. However, to date, the inherent complexity of carbohydrate synthesis has prevented the production of pEtN cellulose oligomers beyond a disaccharide[26].

Here, we report the synthesis by automated glycan assembly[27] (AGA) of nine pEtN cellulose oligosaccharides with varying chain lengths, degrees and patterns of pEtN substitution. The interaction of these glycans with a representative amylogenic peptide of curli (R5)[28,29] is studied. Co-assembly experiments generate artificial fibers and matrices with morphologies and mechanical properties depending on the oligosaccharide structure. Unnatural, synthetic oligosaccharides disrupt or modulate the artificial fibers. These results suggest that selective polysaccharide modification is a valuable approach to generate tunable biofilm-inspired materials.

## Results

**Synthesis of pEtN-substituted oligosaccharides.** The pEtN-substituted oligosaccharides were prepared by a combination of AGA and post-AGA steps. The cellulose backbone was constructed by AGA, following cycles of glycosylation and Fmoc deprotection on solid support L1 (Fig. 1b). **BB1** allowed for linear chain elongation. **BB2** was designed with a levulinoyl (Lev) ester at C-6 that can be selectively hydrolyzed to unmask the hydroxyl group for the subsequent introduction of pEtN. **BB1** and **BB2** were strategically assembled to generate oligomers with the desired pattern of hydroxyl groups. **BB3** was employed in the last cycle of the assembly. After Lev removal, the oligosaccharide backbone was cleaved from the solid support and subjected to post-AGA transformations. The available hydroxyl groups were coupled to the H-phosphonate **4** to give the protected phosphorylated compounds, upon oxidation with aqueous iodine[30]. Steric hindrance made multi-phosphorylation progressively more difficult, requiring five equiv. of **4** and pivaloyl chloride (PivCl) per hydroxyl group to reach full conversion. Excess reagents necessitated extensive purifications to avoid interference with the deprotection steps. Removal of all the remaining protecting groups (PGs) via methanolysis and hydrogenolysis required a careful optimization of the reaction conditions to avoid aggregation/precipitation of the amphiphilic intermediates[31]. Nine zwitterionic compounds were prepared; a mono- and a disaccharide bearing one pEtN group, a trisaccharide carrying two pEtN groups, and six hexasaccharides substituted with one, two or three pEtN units (Fig. 1c). The neutral cellulose analogue **A6** was synthesized as a control.

**Assembly of artificial amyloid fibers.** Well-defined pEtN oligosaccharides provided the bases for exploring the role of the carbohydrate component in biofilm-inspired assemblies. We envisioned an artificial model system consisting of synthetic molecules representatives of the major components of the E. coli ECM. As epitome for the protein part, we selected **R5** (Fig. 1d), the most amyloidogenic repeat of the CsgA unit of curli[28] (the detailed solid-phase synthesis is available in the SI). To generate artificial curli fibers, **R5** was dissolved in hexafluoroisopropanol (HFIP)[32,33]. HFIP was then removed under nitrogen purging followed by evaporation under high vacuum. Addition of water triggered a structural transition from an alpha helix to a beta-sheet conformation, as confirmed by circular dichroism (CD) spectroscopy (Supplementary Fig. 50). The transition was

completed within 20 min (Fig. 2a and Supplementary Fig. 53). Microscopic analysis (AFM, TEM, and SEM, Fig. 2a and Supplementary Fig. 56) performed after 1 or 5 days of incubation showed the presence of ill-defined aggregates.

We then repeated the same assembly process in the presence of the respective oligosaccharide using a **R5**:oligosaccharide mass ratio of 6:1 that best resembles the ECM produced by the uropathogenic E. coli strain UTI89[16]. First, we screened the effect of the oligosaccharide length on the aggregation of **R5**. Co-assembly with the shorter analogues **P**, **PA**, and **PAP** did not significantly affect the structural transition rate of **R5**, while in the presence of longer hexasaccharides, the secondary structure transition of **R5** to the beta-sheet conformation was slower (Fig. 2a and Supplementary Fig. 52). The beta-sheet motif was confirmed by the ThT binding test[34] (Supplementary Fig. 60). Fiber-like structures instead of ill-defined aggregates were detected. We observed a length dependent behavior, with fibers becoming longer with the oligosaccharide chain length.

We then examined seven cellulose hexasaccharides with different degree and pattern of pEtN substitution on the assembly of **R5** (Fig. 2b and Supplementary Figs. 52 and 55). CD analysis indicated that differences in pEtN substitution (degree and pattern) affect the secondary structure transition rate of **R5**, with the sample prepared in the presence of **(PA)3** showing the slowest transition (>6 h) into the beta-sheet conformation (Fig. 2a). Microscopy analysis showed that the **R5** sample containing the unsubstituted cellulose oligomer **A6** assembled into thin fibrils (Fig. 2b, Day 1) that developed into a fibrous network within 5 days (Fig. 2b). All pEtN substituted hexasaccharides also generated fibrils, albeit with different growth rates and morphologies (Fig. 2b, and Supplementary Figs. 52 and 55). While the samples containing the three-substituted oligomers **(PA)3** and **P2APA2** showed long and defined fibrils already on Day 1 (Fig. 2b), the less substituted analogues formed shorter aggregates (Fig. 2b and Supplementary Fig. 55). Interestingly, the fibers observed for **R5** and **(PA)3** adopt the classical curled shape responsible for the name of the natural analogue (Fig. 2b)[35]. On Day 5, all samples formed fibrous networks (Fig. 2b, Supplementary Fig. 55).

The modular approach allowed us to explore different peptide:carbohydrate ratios to better mimic the ECM produced by other bacteria strains. For example, the E. coli AR3110 strain produces a ECM with a much higher pEtN cellulose content (3:1 by mass)[23]. No drastic differences were observed in the fiber morphology, however the fibrils obtained starting from a 3:1 ratio of **R5** and **(PA)3** or **P2APA2** were embedded in a much thicker surrounding matrix (Supplementary Fig. 59b–c). This observation suggests that the pEtN-modified cellulose forms the network connecting the peptide-based fibers, consistent with the existing descriptions of pEtN-cellulose as the "glue" that provides cohesion[8,16].

**Structural analysis.** The fibres obtained from the **R5** sample in the presence of **A6**, **(PA)3** or **P2APA2** showed a similar z-height of around 0.8 nm (Fig. 3a), suggesting that the fibrils are built on the same peptide core. The sample containing **R5** and **A6** showed "naked" fibers together with random aggregates, identified as self-sorted **A6** clusters with height of 4 nm (Fig. 2b and Supplementary Fig. 57, indicated with white arrows and corresponding to Supplementary Fig. 58). In contrast, the fibers generated from the sample containing **R5** and **(PA)3** or **P2APA2** were embedded in a thin matrix (Fig. 3a middle and bottom, highlighted with white arrows). Non-stained TEM images confirmed the presence of a matrix around the fibrils, showing the fibers brighter than the surrounding[36] (Fig. 2b).

## a  Screening of oligosaccharides with different lengths

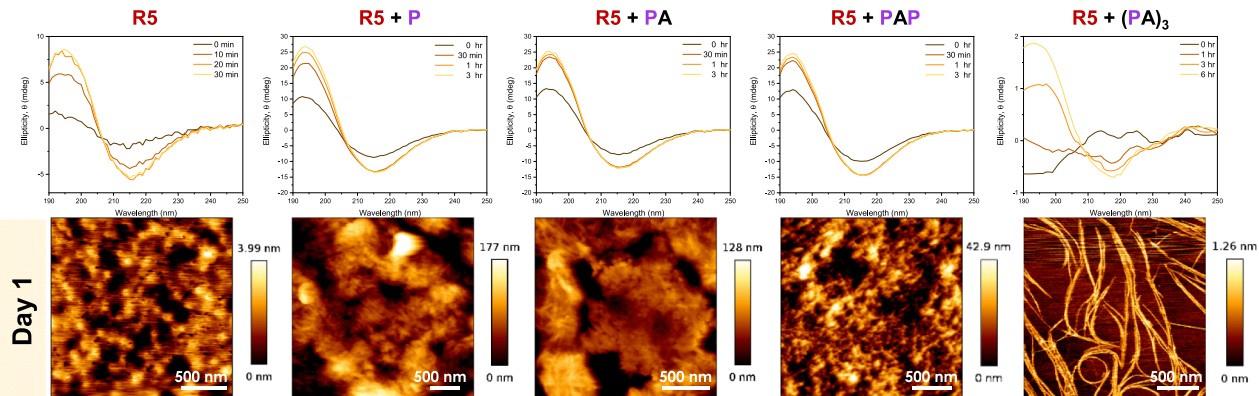

## b Screening of hexasaccharides with different degrees and patterns of pEtN substitution

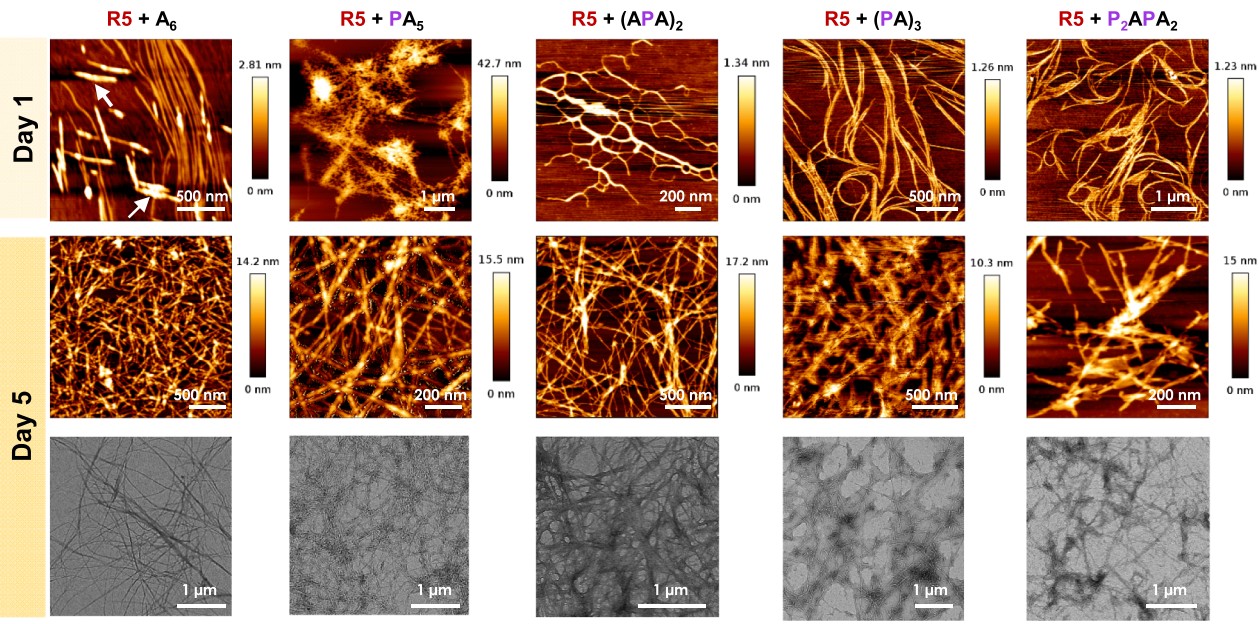

**Fig. 2 Assembly of R5 in the presence of selected oligosaccharides. a** Screening of oligosaccharides with different lengths. CD spectra and AFM (Day 1) of the samples containing **R5** alone or in the presence of different oligosaccharides. The **R5** alone sample results in ill-defined aggregation, whereas fiber-like structures of different dimensions are generated when **R5** is assembled in the presence of an oligosaccharide. **b** Screening of hexasaccharides with different degrees and patterns of pEtN substitution. AFM (Day 1 and Day 5) and TEM (Day 5) of the samples containing **R5** in the presence of hexasaccharides with different degrees and patterns of pEtN substitution. The aggregation of **A₆** is indicated with white arrows.

To gain insights into the molecular interaction between **R5** and the oligosaccharides, we employed solution-state NMR spectroscopy following an approach that revealed key interactions between synthetic heparin oligosaccharides and amyloid fibers[37]. 2D $^1$H-$^1$H total correlated spectroscopy (TOCSY) helped the assignment of the nineteen amidic protons of **R5** (Supplementary Fig. 63). This sample suffered from poor solubility due to aggregation, as shown by the broadening and decreased intensity of the NMR signal with time (Supplementary Fig. 64). The three samples containing both **R5** and **A₆**, **(PA)₃** and **P₂APA₂** respectively showed higher solubility and chemical shift perturbations for selected amino acids (Fig. 3b). Tyrosine, glutamine, histidine and serine were the most affected amino acids in all three samples, albeit to a different extent (Fig. 3b, top panels). The $^{31}$P-NMR signals of the pEtN groups did not show any significant

line broadening or chemical shift perturbation (Supplementary Fig. 72), indicating that the pEtN groups are not directly involved in the interaction with **R5**[38]. Taken together, these results indicate that the presence of the oligosaccharide slow down the **R5** transition into the beta-sheet conformation, favoring the formation of long amyloid fibers over ill-defined aggregates[39,40]. This could be the result of a direct peptide-oligosaccharide interaction that leaves the ionic pEtN groups exposed to water or of a change in the peptide environment due to the presence of the oligosaccharide.

**Mechanical properties of artificial biofilm-inspired matrices.** The co-assembled samples that generated fibers were drop-casted on a glass slide to prepare artificial biofilm-inspired matrices with a thickness of around 300 nm. Their mechanical

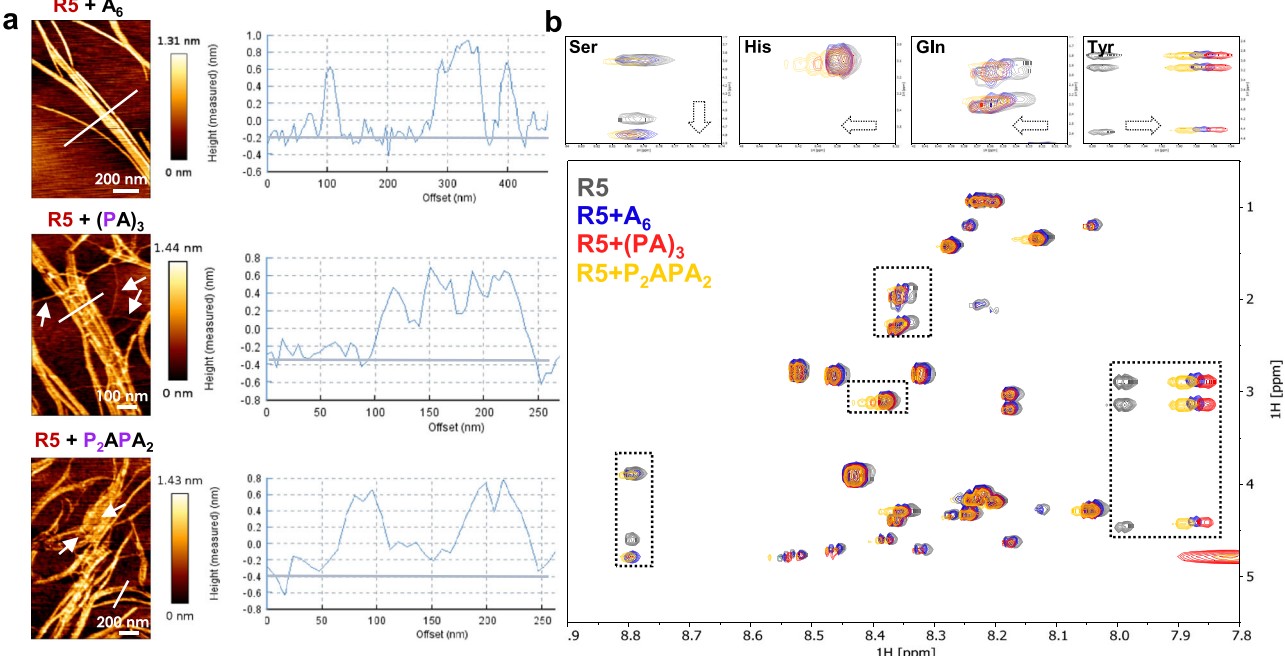

**Fig. 3 Structural analysis of the fibers generated from the assembly of R5 in the presence of three different oligosaccharides. a** AFM images of the fibers at Day 1. The average fiber height is around 0.8 nm for all three samples. The enmeshed matrices are highlighted with white arrows. **b** Overlay of a selected region of the $^1$H-$^1$H TOCSY spectra for the samples containing **R5** alone (gray) and in the presence of **A$_6$** (blue), **(PA)$_3$** (red) and **P$_2$APA$_2$** (yellow). Each spectrum was recorded with a **R5** concentration of 200 μM in $H_2O/D_2O$ (9:1) at 25 °C. The four amide protons mostly affected by the presence of the oligosaccharides are highlighted (top panels), showing a change in chemical shift (His, Gln, Tyr) or signal broadening (Ser).

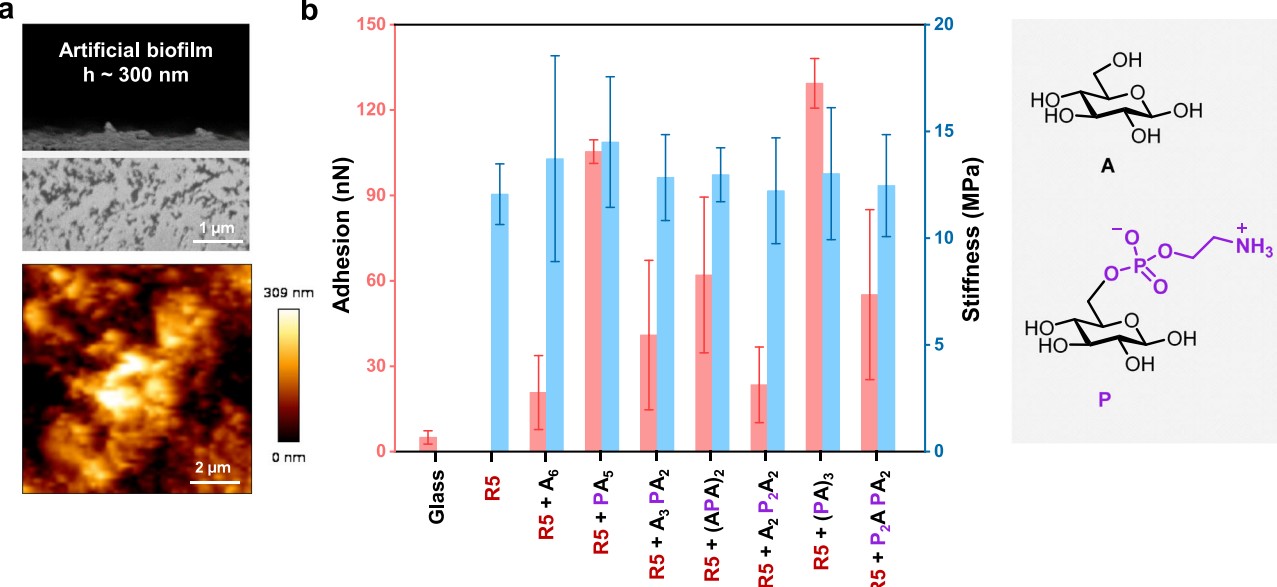

**Fig. 4 Mechanical properties of artificial matrices generated from the assembly of R5 in the presence of different hexasaccharides. a** A cross-sectional SEM and AFM image of the film of **R5** with **(PA)$_3$**. **b** Adhesions and stiffness resulting from AFM force-distance curves (see Supplementary Figs. 73, 74 and Table 17). Each data point corresponds to the mean of 50–100 force measurement and the error bars represent the standard deviation of the mean. Adhesion for **R5** only could not be measured due to inhomogeneity of the film.

properties were explored using AFM force-distance curve analysis (Fig. 4). A stiffness of around 12 MPa for all the matrices was measured with AFM nanoindentation experiments, indicating that the peptide fibres are the major structural component of the artificial matrix. The presence of the pEtN-modified oligosaccharides dramatically enhanced adhesion. The adhesion force for the sample containing **R5** and **(PA)$_3$** was around 130 nN, six times higher than the value obtained for the

sample containing **R5** and **A$_6$**. No direct correlation between the number of pEtN groups and the adhesion was found. The highest values were measured for compounds with the pEtN moiety coupled to the non-reducing end glucose (i.e., **(PA)$_3$** and **PA$_5$**). Multiple pEtN substituents in close vicinity (e.g., **A$_2$P$_2$A$_2$**) resulted in much lower adhesion forces, underscoring the importance of the substitution pattern in determining the mechanical properties of the film.

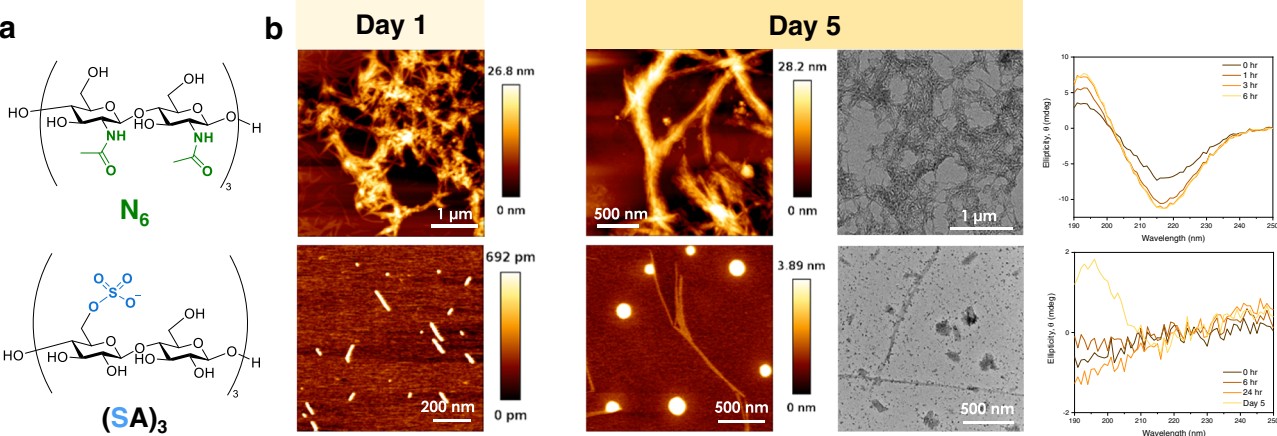

**Fig. 5 Exploring the effect of unnatural oligosaccharides on the assembly of R5. a** Chemical structure of *N*-acetyl glucosamine hexasaccharide **N₆** and sulfated hexasaccharide **(SA)₃**. **b** AFM (Day 1 and Day 5), TEM (Day 5) images, and CD spectra of fibrils prepared with **R5** in the presence of **N₆** or **(SA)₃**.

**The effect of unnatural oligosaccharide modifications.** The discovery of the naturally modified pEtN-cellulose opened up opportunities to generate tuneable materials upon engineering of the carbohydrate components[18]. It has been shown that carbohydrates can modulate the formation of neurotoxic amylogenic fibrils, with chitosan oligosaccharides inhibiting aggregation[41] and heparan sulfates promoting fiber formation[42]. To explore the effect of different glycan modifications on **R5** aggregation, two hexasaccharides not present in natural bacterial biofilms were prepared following established protocols[43,44] (Fig. 5a). **N₆** is a neutral analogue of **A₆** that carries an acetyl amino substituent in position C-2. **(SA)₃** is an analogue of **(PA)₃** in which the pEtN groups are replaced by negatively charged sulfate moieties. In the presence of the *N*-acetyl glucosamine hexasaccharide **N₆**, the secondary structure transition of **R5** into beta-sheet was completed in less than 3 h (Fig. 5b). Fibrils shorter than 1 µm that further aggregated into supramolecular bundles were formed (Fig. 5b and Supplementary Fig. 56). Artificial matrices composed of **R5** and **N₆** were prepared, showing comparable stiffness but higher adhesion than the samples prepared from **R5** and **A₆** (Supplementary Table 17). In contrast, the negatively charged sulfated hexasaccharide **(SA)₃** interrupted the **R5** transition into the beta-sheet conformation (Fig. 5b) and the formation of fibrils (Fig. 5b and Supplementary Fig. 56). This inhibition might be a consequence of strong columbic interactions between the negatively charged oligosaccharide **(SA)₃** and the cationic groups on **R5**, stressing the importance of the zwitterionic pEtN groups in directing **R5** aggregation. The ability of the sulfated hexasaccharide, **(SA)₃**, to inhibit amyloid formation renders this compound an interesting starting point for novel approaches toward the treatment of neurological diseases or as antibacterial agent[45,46].

## Discussion

A collection of pEtN-modified oligosaccharides was synthesized with full control over chain length, degree and pattern of substitutions, providing essential standards to study complex biological systems. The oligosaccharides were incubated with a synthetic peptide, **R5**, representative of curli, to generate a modular model of the *E. coli* biofilm ECM and break down its complexity. Full control over the chemical structure of the individual components permitted to explore the role of oligosaccharide length and substitution in peptide aggregation. While shorter oligomers had little effect, the longer hexasaccharides slowed down the secondary structure transition into beta-sheet of the peptide **R5**, inducing the growth of extended fibrous structures. The oligosaccharide fine structure dramatically affected the fiber growth rate and the mechanical properties of the composite.

We demonstrated that not only the degree, but also the pattern of pEtN substitution influences adhesion. In contrast, stiffness remains unchanged for all samples indicating its strong connection to the peptide component. Modifications beyond the natural one were screened, delivering interesting targets for the future production of engineered biofilm-inspired materials. Metabolic engineering[47] and/or directed evolution approaches[48,49] may introduce such modifications in vivo and produce novel cellulosic materials with non-natural modifications.

## Methods

**Synthesis.** The oligosaccharides were prepared using a home-built synthesizer designed at the Max Planck Institute of Colloids and Interfaces. The solid-phase peptides synthesis (SPPS) of **R5** was performed with a microwave-assisted peptide synthesizer (Liberty Blue, CEM, USA). All details concerning building block synthesis, AGA modules, post-AGA manipulations, and SPPS can be found in Supplementary Information.

**Assembly of artificial fibers and matrices.** Stock solutions were prepared dissolving separately **R5** and the oligosaccharides in HFIP with a concentration of 200 µM (0.4 mg mL⁻¹) and 0.13 mg mL⁻¹, respectively. The **R5** and oligosaccharide stock solutions were mixed with 2 to 1 (or 1 to 1) volume ratio to reach the final mass ratio with 6 to 1 (or 3 to 1) and sonicated for 10 min. HFIP was removed under gentle nitrogen purging followed by evaporation under high vacuum. Water was added to the dried films to reach the final peptide concentration of 25 µM for imaging, CD, and ThT binding test, and 200 µM for 2D TOCSY NMR analysis. The artificial films were prepared by drop-casting of a 25 µM solution of the co-assembled sample on a pre-washed glass substrate to generate films with a thickness of 300 nm. The co-assembled sample were prepared with a 6:1 peptide:oligosaccharide mass ratio and incubated for 5 days before drop-casting. AFM imaging and force measurement were performed in air in an AFM chamber with a relative humidity of 25%. If not mentioned, the standard ratio between **R5** and oligosaccharide is 6 to 1 by mass. All details concerning fibrils' structural analysis and films' mechanical properties can be found in Supplementary Information.

## Data availability

The authors declare that all data supporting the findings of this study are available within the paper and in the Supplementary Information files. Data are also available from the corresponding author upon request.

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

## Acknowledgements

We thank the Max Planck Society, the MPG-FhG Cooperation Project Glyco3Dysplay, and the German Federal Ministry of Education and Research (BMBF, grant number 13XP5114) for generous financial support. J.Y.H. acknowledges the International Max Planck Research School on Multiscale Bio-Systems for funding. J.Y.H. and M.D. acknowledge support from the Max Planck Queensland Centre on the Materials Science for Extracellular Matrices. D.V.S. and S.P.M. thank the RIKEN-Max Planck Joint Center for Systems Chemical Biology for financial support. We thank Dr. Cécile Bidan for proofreading the paper.

## Author contributions

T.T.E. and M.D. conceived this project. T.T.E. and J.Y.H. performed the synthesis and the NMR analysis. S.G. developed the assembly methods, performed microscopic measurements and the mechanical property characterization of the films, and analyzed the data. J.Y.H. performed the nano-indentation of the films and analyzed the data. S.P. assisted with the synthesis of **R5**. M.D. supervised the project. All authors contributed to and discussed the paper.

## Funding

## Competing interests

The authors declare no competing interests.
