## [Peer Review File · Nature Communications]

Synthetic phosphoethanolamine-modified oligosaccharides reveal the importance of glycan length and substitution in biofilm-inspired assembliesEditorial Note: This manuscript has been previously reviewed at another journal that is not operating a transparent peer review scheme. This document only contains reviewer comments and rebuttal letters for versions considered at Nature Communications. Mentions of the other journal have been redacted.

REVIEWERS' COMMENTS

Reviewer #3 (Remarks to the Author):

As I am re-reviewing this manuscript I'll cut to the chase: the authors have provided satisfactory edits and answers to my concerns. This is a very interesting system and I am impressed by the tunability of these systems. I would be very excited to see how these systems interact with bacteria and epithelial cells, but those experiments are very much outside the scope of this paper. I think this paper should be accepted as is.

Reviewer #4 (Remarks to the Author):

My revision of the manuscript authored by Tyrikos-Ergas et al. and entitled "Synthetic phosphoethanolamine-modified oligosaccharides reveal the importance of glycan length and substitution in biofilm-inspired assemblies" was focused on the appropriateness of the authors' response to the comments of Reviewer 1.

The authors claim (see the Abstract, text and Conclusions paragraph) that the major novelty of their work is "that the pattern of pEtN modification drastically affects the mechanical properties of synthetic biofilm-inspired matrices". This observation is based on the study carried out using synthetic cellulose oligosaccharides (in particular hexasaccharides, but also shorter oligosaccharides and "unnatural" hexasaccharides were screened) modified with phosphoethanolamine at different 6-OH, providing different modification patterns. A second major claim is the development of a matrix model including synthetic oligosaccharides and peptides to determine how they affect peptide aggregation depending on their modification pattern. In my opinion, this represents a significant progress beyond the state of the art of the seminal discovery of Cegelsky, who showed that phosphoethanolamine-modified cellulose produced by *E. coli* governs bacterial adhesion. The authors third claim concerning the "very first synthesis" is in my opinion less relevant. Considering the high impact of the journal, a solid and well-established, but not innovative synthetic approach was employed.

I do not consider significant the lack of biological study with living cells: I agree with the authors that this is out of scope, the manuscript contains sufficient informations at the current stage. For the same reason (lack of detailed biological studies), I do not think the work is suitable for [REDACTED], but it could be appropriate for Nature Comm.

Other minor remarks concerning figures and typos have been fully addressed.

Overall, in my opinion the manuscript may deserve publication [REDACTED] on Nature Comm.

Reviewer #5 (Remarks to the Author):

The authors addressed Reviewer 2's concerns. The changes throughout the manuscript to specify that a synthetic biofilm-inspired material was studied cleared up many of the concerns that some of

the text/emphasis in the previous version focused on biofilms despite that biofilms were never grown or bacterial cells used in studies. Additionally, I feel that the authors' statements around using R5 as a model system adequately addressed concerns that R5 is more amylogenic than full CsgA.

Below please find a detailed response to all the reviewers' comments.

Reviewer #3:

Comment: As I am re-reviewing this manuscript I'll cut to the chase: the authors have provided satisfactory edits and answers to my concerns. This is a very interesting system and I am impressed by the tunability of these systems. I would be very excited to see how these systems interact with bacteria and epithelial cells, but those experiments are very much outside the scope of this paper.

I think this paper should be accepted as is.

Response: We thank the reviewer for the time invested in analyzing our paper and for the useful suggestions. We are equally excited to learn how our synthetic system will behave in more complex biological environments.

Reviewer #4:

Comment: My revision of the manuscript authored by Tyrikos-Ergas et al. and entitled "Synthetic phosphoethanolamine-modified oligosaccharides reveal the importance of glycan length and substitution in biofilm-inspired assemblies" was focused on the appropriateness of the authors' response to the comments of Reviewer 1.

The authors claim (see the Abstract, text and Conclusions paragraph) that the major novelty of their work is "that the pattern of pEtN modification drastically affects the mechanical properties of synthetic biofilm-inspired matrices". This observation is based on the study carried out using synthetic cellulose oligosaccharides (in particular hexasaccharides, but also shorter oligosaccharides and "unnatural" hexasaccharides were screened) modified with phosphoethanolamine at different 6-OH, providing different modification patterns. A second major claim is the development of a matrix model including synthetic oligosaccharides and peptides to determine how they affect peptide aggregation depending on their modification pattern. In my opinion, this represents a significant progress beyond the state of the art of the seminal discovery of Cegelsky, who showed that phosphoethanolamine-modified cellulose produced by *E. coli* governs bacterial adhesion. The authors third claim concerning the "very first synthesis" is in my opinion less relevant. Considering the high impact of the journal, a solid and well-established, but not innovative synthetic approach was employed.

I do not consider significant the lack of biological study with living cells: I agree with the authors that this is out of scope, the manuscript contains sufficient informations at the current stage. For the same reason (lack of detailed biological studies), I do not think the work is suitable for [REDACTED] but it could be appropriate for Nature Comm.

Other minor remarks concerning figures and typos have been fully addressed.

Overall, in my opinion the manuscript may deserve publication [REDACTED] on Nature Comm.

Response: We thank the reviewer for the time invested in analyzing our paper. We agree that the synthesis alone is not the main novelty of the manuscript, but provides the tools to study more complex biological systems. This concept is expressed in the conclusion of our paper: “A collection of pEtN-modified oligosaccharides was synthesized with full control over chain length, degree and pattern of substitutions, providing essential standards to study complex biological systems.”

Reviewer #5:

Comment: The authors addressed Reviewer 2's concerns. The changes throughout the manuscript to specify that a synthetic biofilm-inspired material was studied cleared up many of the concerns that some of the text/emphasis in the previous version focused on biofilms despite that biofilms were never grown or bacterial cells used in studies. Additionally, I feel that the authors' statements around using R5 as a model system adequately addressed concerns that R5 is more amylogenic than full CsgA.

Response: We thank the reviewer for the time invested in analyzing our paper.